# Approximate Bayesian computation with deep learning supports a third archaic introgression in Asia and Oceania

Mayukh Mondal [1], Jaume Bertranpetit [2] & Oscar Lao [3,4]

Since anatomically modern humans dispersed Out of Africa, the evolutionary history of Eurasian populations has been marked by introgressions from presently extinct hominins. Some of these introgressions have been identified using sequenced ancient genomes (Neanderthal and Denisova). Other introgressions have been proposed for still unidentified groups using the genetic diversity present in current human populations. We built a demographic model based on deep learning in an Approximate Bayesian Computation framework to infer the evolutionary history of Eurasian populations including past introgression events in Out of Africa populations fitting the current genetic evidence. In addition to the reported Neanderthal and Denisovan introgressions, our results support a third introgression in all Asian and Oceanian populations from an archaic population. This population is either related to the Neanderthal-Denisova clade or diverged early from the Denisova lineage. We propose the use of deep learning methods for clarifying situations with high complexity in evolutionary genomics.

[1] Institute of Genomics, University of Tartu, Riia 23b Tartu, Tartu 51010, Estonia. [2] Institute of Evolutionary Biology (CSIC-UPF), Universitat Pompeu Fabra, Doctor Aiguader 88 (PRBB), 08003 Barcelona, Catalonia, Spain. [3] CNAG-CRG, Centre for Genomic Regulation (CRG), Barcelona Institute of Science and Technology (BIST), Baldiri i Reixach 4, 08028 Barcelona, Spain. [4] Universitat Pompeu Fabra (UPF), Barcelona, 08003 Catalonia, Spain. Correspondence and requests for materials should be addressed to J.B. (email: jaume.bertranpetit@upf.edu) or to O.L. (email: oscar.lao@cnag.crg.eu)

All modern humans are genetically related to each other at a time depth of up to 300 thousand years ago (Kya)[1,2] and share a common African root[3–5]. The migratory routes used by anatomically modern humans (AMH) after the African diaspora and aspects of the interbreeding between AMH and presently extinct hominins living at the time in Eurasia (here referred as Eurasian Extinct Hominins, EEH) are still under debate. Recently, several genetic studies have argued that there was only one main Out of Africa (OOA) event[6–8] that happened less than 100 Kya, with earlier dispersals, which most likely does not contribute much to present-day variation[9–11]. Sequencing of ancient Neanderthal and Denisovan fossils supported introgression events into AMH out of Africa;[12,13] however, recent studies also support the presence of gene flow from AMH into Neanderthals[14,15], thus suggesting a complex hominin evolution.

A complex demographic model considering archaic introgressions out of Africa is needed to properly address two main points. First, the number of introgression events from Neanderthal and Denisova into AMH, which remains controversial, and their impact on the AMH gene pool. A difficult point is to account for the higher amount of Neanderthal introgression in Asia than in Europe[8]. Moreover, the geographic limits of Denisovan introgression are not well established[14,16]. Interestingly, Australian Aborigines also harbour high amounts of EEH due to several introgression events[7,9,12]. Although the evidence for Denisovan introgression in Oceanian populations is clear, the amount of introgression varies between 3 and 6% depending on the methodology[14,16,17]. Also, there is disagreement about the number of introgression events necessary to explain patterns of admixture in modern Australian Aborigines[7,8,11]. The second main point is to identify all the cryptic ghost archaic populations that have introgressed with hominins with known genomes (modern humans from the Old World, Neanderthals and Denisovans). It has been suggested that other archaic hominins could have interbred with Denisovans[14,16]. Furthermore, introgression of extinct hominins beyond those of Neanderthals and Denisovans into some AMH populations, both within Africa[2,18], as well as in South Asia[6] and Tibetan populations[19] has been proposed. Recently it has been also found that there could have existed two different pulses of Denisovan admixture in East Asia[20].

However, comparing the likelihood of competing complex demographic models given the observed data is cumbersome[21], requires manual expertise[22] and/or fitting[23]. Consequently, all of these different demographic models have not been tested together using the same methodology. Furthermore, it would be interesting to have an estimation of the parameters that define a demographic model given the data. However, this is not available in commonly applied algorithms (i.e. Admixtools[23]). Within this context, Approximate Bayesian Computation (ABC) is a flexible statistical framework that allows estimating the posterior distribution of a parameter/model through the generation of simulated datasets for cases when there is no close-form of the likelihood of the function[24]. ABC is widely used in the field of population genetics[25]. However, identifying the proper transformations of the data to generate informative summary statistics (SS) is model and parameter dependent, and including non-informative redundant SS can jeopardize the performance of the ABC approach[22]. Jiang et al.[26] recently proposed a theoretical improvement of the ABC framework by considering Deep Learning (DL) for generating informative SS from a set of (raw) SS. DL is a type of Artificial Neural Network (ANN) topology based on multiple hidden neural layers, and it has been used for population genetics inference in non-ABC frameworks[27]. Since ANN performs a nonlinear transformation of the input features towards the output[28], DL provides an optimal way of extracting and maximizing the non-redundant information out of the raw

SS[26]. Thus, a DL can be trained using a large number of raw SS as input features to predict the parameters that generated the simulated data, or to classify a simulated data in a set of models, and use this classification prediction or parameter estimation as a new SS (SS-DL).

In the present study, we considered the multidimensional absolute site frequency spectrum (SFS) as raw SS. SFS accounts for the number of single nucleotide polymorphisms (SNPs) present in a particular combination of derived alleles in the sampled populations[29]. This statistic is the core of many frequency based statistics applied in population genetics for distinguishing among competing topological trees and detecting archaic introgression (i.e. D-statistics, F4, F4 ratio etc[23].), determining migration rates, time of split and/or differences in the effective population sizes in models of two populations that split and migrate (i.e. Fst[29]). Moreover, the SFS is not affected by recombination[30]. SFS has been shown to allow robust demographic estimates[21,29,31] and the folded SFS has been previously used in classical ABC frameworks for estimating the population size[32]. However, SFS is high dimensional and it combinatorically increases on the considered number of samples and populations[29]. Our proposed methodology takes advantage of the nonlinear nature of the DL for compressing the SFS to extract the underlying patterns defined by all the dimensions of the SFS that allow distinguishing among proposed models or obtaining the value of the demographic parameters of interest. This SS-DL is then used in an ABC framework.

We apply our newly developed method in a large set of whole genome sequences of present and ancient remains to estimate the posterior probability of competing complex demographic models with a focus on introgression events. We find that ABC-DL is powerful for distinguishing between plausible introgression models and propose a third archaic introgression event happened (in addition to Neanderthal introgression to all OOA populations and Denisova introgression for Oceanian population), which is common to all Asia and Oceanian populations.

## Results

**D-statistics and F4 ratio analysis.** D-statistics and F4 ratio statistics, tests of frequent use in population genetics, have consistently demonstrated the existence of Neanderthal and Denisovan ancestry in out of African populations[16]. Using a common variant calling (see Methods and also Supplementary Methods), we found with D-statistics and F4 ratio test that all Asian populations (East Asians, Andamanese and Indian Tribal populations) have higher amounts of Neanderthal ( >1%) and Denisovan ancestry ( >1%) compared to Europeans (Table 1). Interestingly, when trying to calculate the excess of Neanderthal ancestry in Asian populations compared to Europeans using the F4 ratio test, different outgroups (ancestral alleles mainly constituted from the chimpanzee reference genomes[33] and Denisovan[12]) gave different results (Fig. 1). This was unexpected, as the F4 ratio test and D-statistics should not be affected by changing the outgroup under the simplistic topology used in these tests. When using the ancestral alleles from the 1000 Genomes as outgroup[33], the excess of Neanderthal ancestry in Asian populations is around 0.5% (s.e. 0.15%) and statistically significant (Z score > 3). In contrast, when using Denisovan as outgroup, this estimate diminishes and loses statistical significance (~0% ± 0.15%). We then used simulations considering different demographic models (Supplementary Figure 1), varying the percentage of introgression while keeping constant all the other parameters (Supplementary Table 3), to fit the observed D-statistics and F4 ratio statistics. The simulation results suggest that a simple model for the excess of Neanderthal and Denisova ancestry in Asian populations is not enough for fitting these statistics and other

**Table 1 D-statistics of Neanderthal and Denisova ancestry and dearth of African Ancestry in non-African populations**

|   | W | X | Y | Z | D score | Z score | Simulation |
|---|---|---|---|---|---|---|---|
| A | AFR | EUR | NEAN | Ancestral | −0.0457 | −11.980 | −0.029 |
|   | AFR | ASN | NEAN | Ancestral | −0.0557 | −12.028 | −0.051 |
|   | AFR | AND | NEAN | Ancestral | −0.0536 | −12.041 | −0.051 |
|   | AFR | IND | NEAN | Ancestral | −0.0512 | −13.131 | −0.051 |
|   | AFR | PAC | NEAN | Ancestral | −0.0753 | −13.797 | −0.066 |
| B | EUR | ASN | NEAN | Ancestral | −0.0126 | −2.428 | −0.034 |
|   | EUR | AND | NEAN | Ancestral | −0.0102 | −2.055 | −0.034 |
|   | EUR | IND | NEAN | Ancestral | −0.0073 | −1.758 | −0.035 |
|   | EUR | PAC | NEAN | Ancestral | −0.0374 | −5.851 | −0.056 |
| C | EUR | ASN | DENI | Ancestral | −0.0096 | −2.663 | −0.033 |
|   | EUR | AND | DENI | Ancestral | −0.0175 | −4.165 | −0.033 |
|   | EUR | IND | DENI | Ancestral | −0.0151 | −3.795 | −0.033 |
|   | EUR | PAC | DENI | Ancestral | −0.0842 | −15.219 | −0.105 |
| D | EUR | ASN | AFR | Ancestral | 0.0015 | 0.661 | −0.011 |
|   | EUR | AND | AFR | Ancestral | 0.0093 | 3.3583 | 0.016 |
|   | EUR | IND | AFR | Ancestral | 0.0064 | 2.557 | 0.017 |
|   | EUR | PAC | AFR | Ancestral | 0.0329 | 10.488 | 0.044 |

Section A shows the introgression amount of Neanderthals to all non-African populations compared to African populations using D-statistics. As African populations do not have introgression from EEH (Eurasian Extinct Hominins) populations, these values can be regarded as absolute amount of introgression. Section B shows the introgression amount of Neanderthals to Asian and Pacific populations compared to European populations. As European populations have introgression from Neanderthal, these values represent the increase of Neanderthal ancestry in Asian and Pacific populations. Section C shows the introgression amount of Denisova to Asian and Pacific populations compared to European populations. As the Neanderthal introgression in European populations can increase the Denisova ancestry calculated by D-statistics because of incomplete lineage sorting, these values represent the increase of Denisova ancestry in Asian and Pacific populations compared to European populations. Section D shows the dearth of African ancestry in Asian and Pacific populations compared to European populations. As introgression from EEH population will increase the dearth of African ancestry, these values are a good proxy for the relative increase amount of Introgression from EEH populations. In the last column, mean of D-scores of 500 replicates using the mean value from the posterior distribution of parameters of model H
*AFR* African, *ASN* East Asian, *IND* Tribal Indian, *AND* Andamanese, *EUR* Europeans, *PAC* Pacific/Oceanian, *NEAN* Neanderthal, *DENI* Denisova

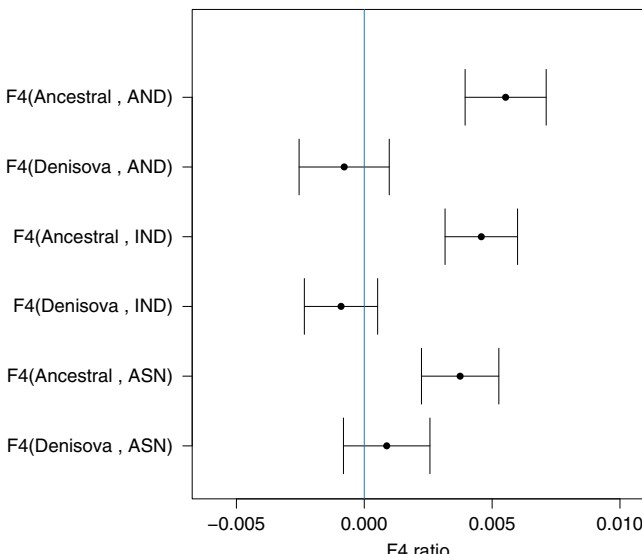

**Fig. 1** Increase of Neanderthal ancestry in Asians compared to Europeans. The introgression amount is calculated by the F4 ratio test. F4 (Outgroup, Altai; Europeans, X)/F4(Outgroup, Altai; Africa, Vindjia) where Outgroup can be either Denisova or Ancestral alleles from 1000 Genome and X is either East Asian (ASN), Indian Tribal (IND) or Andamanese (AND). In the y axis to make it short we only mention F4(Outgroup, X) as the other populations remain constant for all the F4 ratio tests. In the x axis we have the amount of F4 ratio and the standard error is denoted by bars (standard error was calculated by Jackknife method using 555 blocks). The blue line signifies 0 value for F4 ratio test

ancestral component is needed (Supplementary Figure 3). Moreover, our analyses suggest that D-statistics and F4 ratio are sensitive to the outgroup (Denisova or chimpanzee) used at least for archaic introgression model. In Oceania, results get more complex to interpret due to the derived allele sharing between Neanderthals and Denisovans. Given the possible biases due to the reference genome (mostly of European origin) used to

calculate D-statistics, we re-mapped all the genomes to the chimpanzee reference and recalculated the results (Supplementary Table 2). Although most of the Neanderthal and Denisovan ancestry calculations remained the same, we note a decrease of African ancestry by changing the reference from human to chimpanzee in Indian Tribal and Oceanian populations. Overall, these results show a complex situation that needs a broader approach to distinguish among competing demographic models.

**ABC-DL analysis**. In order to quantify the posterior probability of different demographic models given the observed genetic diversity in current populations and archaic individuals, which cannot be achieved by the D-statistics and F4 ratio analyses, we applied an ABC-DL analysis (see Methods and Supplementary Note 2).

We considered several plausible competing demographic models which describe scenarios that have been proposed by different studies to explain the observed genetic diversity of introgression patterns among current old world AMH populations, Neanderthals and Denisovans (see Fig. 2 and Supplementary Table 5). The implemented demographic models are variations from an initial model with many accepted features (Model A): OOA origin of modern humans, with a Eurasian split between Europeans and the group comprising two subgroups, East Asians, Indian and Andamanese on one hand, and Papuans and Australians on the other. Introgression of Neanderthals is found in all OOA populations, and that of Denisovans in the Oceanian populations. Introgression from an extinct group of distant hominins into Denisovans[14,16] has been considered in all models, and this introgression does not affect directly our core results. From this basic model, several variations have been proposed to better fit the genetic data; here we consider the following. Model B: decrease of European Neanderthal ancestry due to admixture with a modern human, "basal Eurasian", a ghost population non-introgressed by Neanderthals ($X_n$)[34]. Model C: two introgression events from Neanderthals, the first affecting all OOA populations and the second only affecting Asia[35,36]. Model D: two introgression events both from Neanderthals and Denisovans, with a

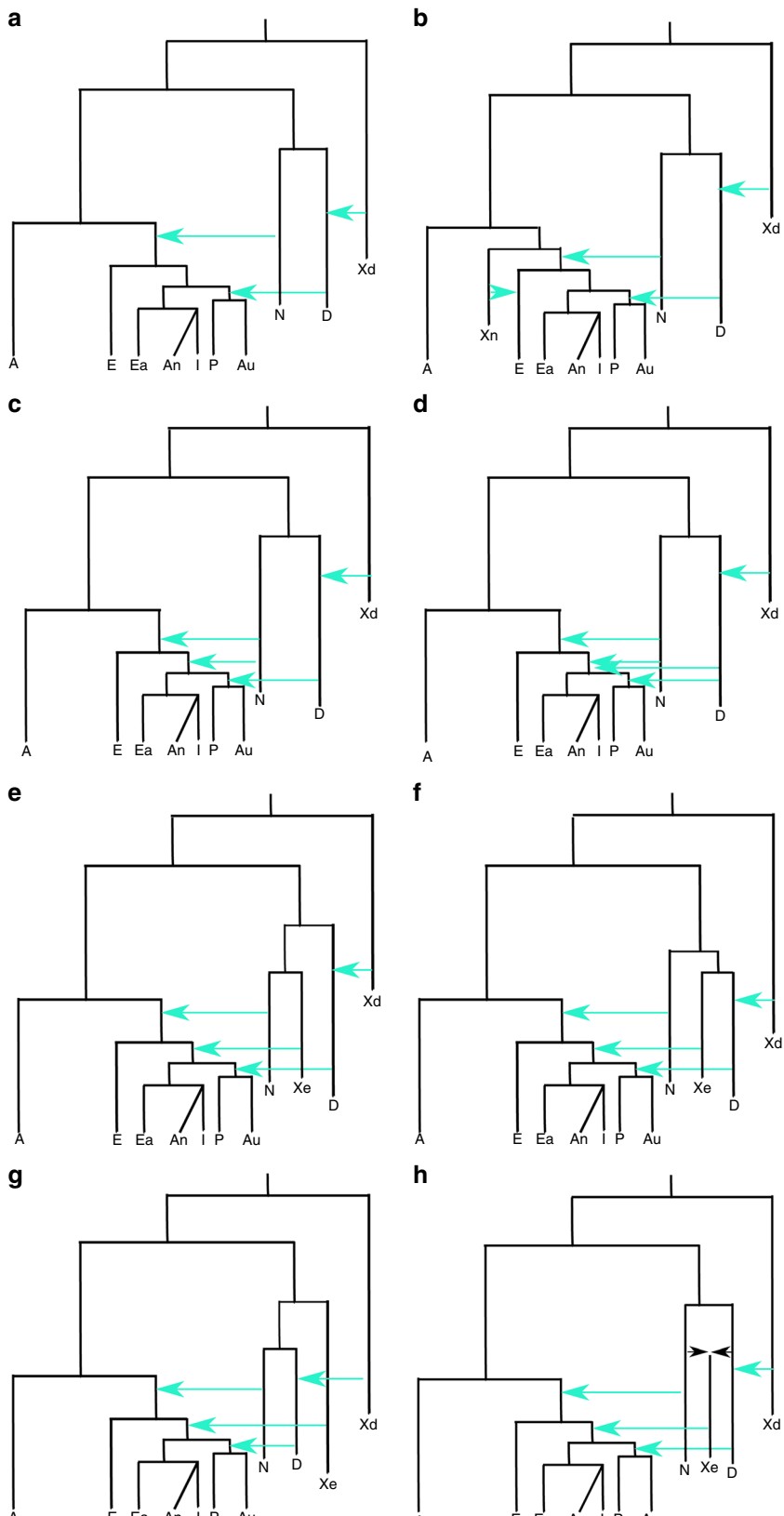

**Fig. 2** Demographic models implemented for explaining the genetic variation present in current African (A), European (E), East Asian (Ea), Andamanese (An), Indian (I), Papuan (P) and Australian (Au) populations, in relation with Altai Neanderthal (N) and Denisovan (D) archaic populations. The process of archaic introgression in Asian populations is modelled either by different introgression events from **a** anatomically modern human ghost population (Xn), **b**, **c**, **d** known archaic populations such as N and D and or **e**, **f**, **g**, **h** by an unknown archaic extinct (Xe) population. An archaic population (Xd) introgressing to Denisovans is also modelled. Turquoise arrows indicate single pulse events of archaic introgression. All models include recent continuous migrations between African and European, European and East Asian, and Papuan and Australian populations

Denisovan introgression on Asians before the split of Oceanian populations[14,16]. The remaining models have a single introgression of Neanderthals and Denisovans (as in Model A) but with an introgression to Asian and Oceanian populations from an extinct (Xe) hominin population that is a sister group of Neanderthals (Model E), or is a sister group of Denisova (Model F;[20]), or is an outgroup of both Neanderthals and Denisovans (Model G;[6]). Given the recently discovered Neanderthal–Denisovan hybrid[37], we also considered a scenario in which this extinct hominin is an admixed population between Neanderthals and Denisovans (Model H)[37], a similar model of a trichotomy as will be seen below. Furthermore, Model E and Model F can be considered particular cases of Model H when the percentage of Denisovan admixture is 0 or 100%, respectively. In all the models, the basic tree structure remained constant although all the demographic parameters—including the amount of introgression- were estimated.

We first evaluated the performance of the proposed ABC-DL methodology for distinguishing between the proposed eight models. We used simulated data as pseudo-observed data and run the full ABC-DL pipeline; we applied a 'hard' model classification to assign the simulated data to the model with the highest posterior probability (see Supplementary Note 2). The confusion matrix (Supplementary Table 6) summarizing the model classification shows that the proposed ABC approach identifies the model that generated the data (Model$_{sim}$) with posterior probability P(Model$_{sim}$ = Model$_{abc}$ | Data) > 50% in all the models. When comparing the eight considered models by ABC-DL with the observed data, the ABC model inference supports model H (P(Model = H|Data) = 0.46), namely the introgression of genes from an archaic Denisovan-Neanderthal admixed extinct population (Xe) into Asian AMH populations, before the split of Oceanian groups, closely followed by Model F (P(Model = F|Data) = 0.38), which models Xe as a sister population of Denisovans. Model H is slightly more likely than Model F (1.2 times more likely), and much more likely than models E (5 times more likely), G (7.6 times) and D (107 times). Models not considering the presence of an extinct archaic 'ghost' population introgressing AMH -A, B, C and D- obtained a posterior probability of ~0 (see Supplementary Figure 4). Therefore, only models with admixture of an extinct population in the Neanderthal–Denisovan clade, but with significant differences to each of them, are supported by the ABC-DL framework.

Next, we implemented ABC-DL to estimate the posterior distribution of each parameter for Model H (Fig. 3b and Supplementary Figure 5). Given that Model F is also similarly supported, for completion we also estimated the posterior distribution of the parameters in this model (Fig. 3a). As internal check of the performance of the DL in Model H, we first estimated the Pearson correlation between the parameter values used for the simulation and the predicted parameter value by the DL (Supplementary Table 7). For almost all the parameters from the Model H, we obtained a high correlation between the real parameter and the predicted one by the DL with the exception of the effective population sizes of the unknown ghost populations, which is a very acceptable expectation as we do not have direct information of the genetic variation in these populations. When applied to the observed data (Table 2 and Supplementary Table 8), our ABC-DL approach for estimating parameters of Model H suggests that (i) the introgression of Neanderthals with AMH out of Africa was 1.3% (CI from 0.18% to 2.5%) and occurred 69 Kya (CI from 56 Kya to 88 Kya; assuming a generation time of 29 years[38]), whereas the split between African and OOA populations took place 121 Kya (CI from 78 Kya to 167 Kya); (ii) the Denisovan introgression to Oceanian populations was 1.6% (CI from 0.4% to 2.5%) and took place 43 Kya (CI from

29 Kya to 50 Kya); finally, (iii) the estimated amount of introgression in the AMH ancestor of current Asian populations by the extinct archaic population Xe was 2.6% (CI from 0.7% to 4.6%) at 51 Kya (CI from 45 Kya to 58 Kya). According to our analyses, Xe appeared 304 Kya (CI from 211 Kya to 375 Kya) from an admixture event of Denisovans and Neanderthals that would have occurred only 14 Kya after the divergence of Denisovans and Neanderthals (314 Kya; CI from 300 Kya to 343 Kya). The estimated divergence between Denisovan and Neanderthals is relatively recent compared to the estimated in[16] (381–473 Kya when considering a mutation rate of 0.5e-09 per bp per year or 1.45e-08 per generation assuming 29 years by generation) but higher than in[16] (190–236 Kya when considering a mutation rate of 1e-09 per bp per year or 2.9e-08 per generation). Since the number of non-shared variants depends on the time of divergence, the genetic drift at each population and the mutation rate, a possible explanation for the differences in time of divergence could be the mutation rate, which substantially variate among studies. Demographic parameters estimated for Model F are similar in magnitude to the ones estimated in Model H, with credible intervals that overlap the ones obtained for Model H (Supplementary Table 9).

In order to see whether the proposed Model H fit observed D-statistics, we generated simulations using the mean of the estimated posterior probability of each parameter, and computed D-statistics assuming different population tree topologies for stressing the impact of archaic introgression in Asian populations. Simulation results from ABC-DL for Model H replicate the empirical results of the D-statistics (Table 1, last column) by reproducing the excess of Neanderthal and Denisova ancestry in Asian populations compared to Europeans.

Taken together, we have an overwhelming support for the existence of a third extinct branch of the Neanderthal–Denisovan clade; considering the credible interval of the time of split between the archaic populations, a model with a trichotomy is a good consensus for Model D-H as in all the models the Xe separates from Neanderthal or Denisova at a similar time depth of Neanderthal and Denisova split (Supplementary Table 10).

## Discussion

We have developed a new methodology for statistically comparing complex demographic models and estimating demographic parameters with focus on applying it to the complex demographic history of introgressions in Eurasia. This methodology takes advantage of the nonlinear capabilities of DL for identifying patterns and extracting features, and of the well-established statistical ABC framework for properly estimating the posterior distribution of models and parameters. Overall, the computational methods used here take full advantage of machine learning technologies through DL to improve the performance of the ABC statistical framework. This new approach overcomes some previous limitations for comparing demographic models, minimizes manual fitting and allows estimating parameters in complex demographic models.

The ABC-DL analysis supports that all Asian and Oceanian populations, including East Asians, received an introgression from an unknown extinct archaic hominin population. However, the exact relationship of this extinct hominin population with the other known archaic populations is not completely disentangled. The two models mostly supported by our ABC-DL model comparison suggest that Xe corresponds to one of the two sister-Denisovan lineages (Model F) or this hominin was closely related to the Neanderthals and Denisovans with a slightly greater similarity to Denisovans than to Neanderthals (Model H). In our analyses, Model H is slightly better supported than Model F.

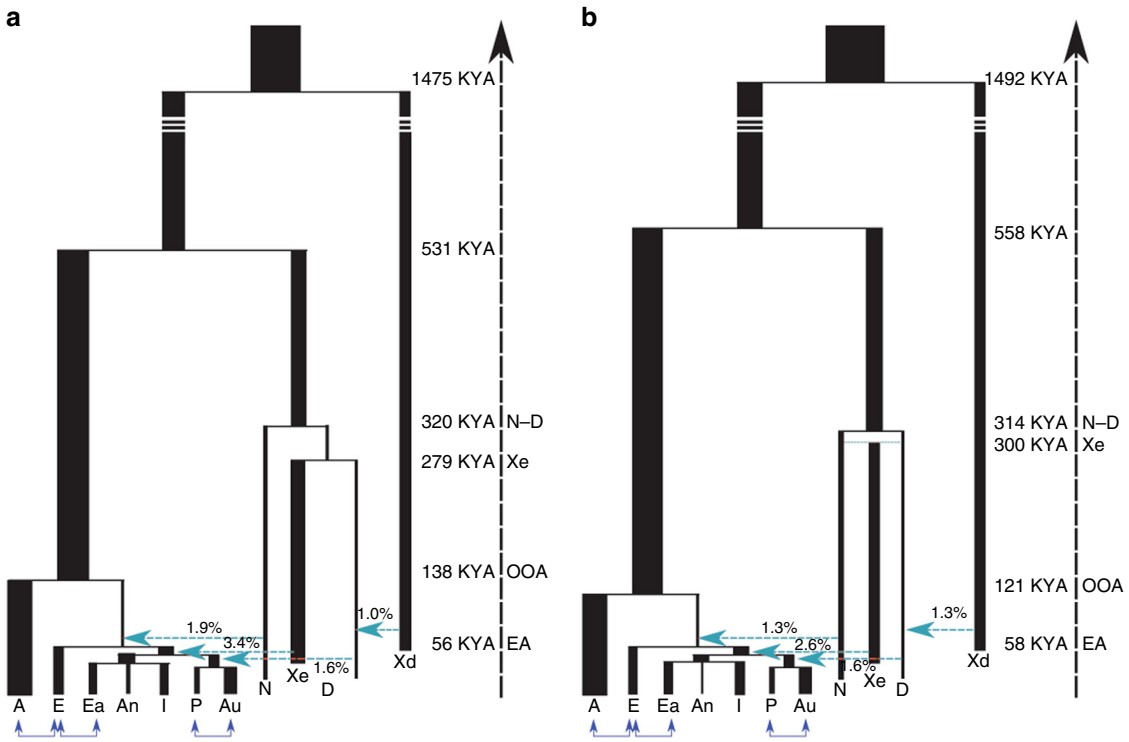

**Fig. 3** Graphical representation of the two best supported model (Model F and Model H) with mean posterior distribution of each parameter. **a** Mean of the posterior distributions of each parameter calculated by means of ABC-DL in model F for Africans (A), Europeans (E), East Asians (Ea), Andamanese (An), Indians (I), Papuans (P), Aboriginal Australians (Au), Altai Neanderthal (N) and Denisovan (D). The model includes an extinct hominin population (Xe) descendent from the Denisovan population. **b** Mean of the posterior distributions of each parameter calculated by means of ABC-DL in model H. Xe population is an admixture between Neanderthals and Denisovans which interbreeds with Asian populations and a *Homo erectus*-like population (Xd) that interbreeds with Denisovan. In blue, some of the recent continuous migrations between populations which were taken into consideration in the model. In turquoise, the different introgressions

| Parameter | Model F | | | Model H | | |
|---|---|---|---|---|---|---|
| | Mean | 2.5% CI | 97.5% CI | Mean | 2.5% CI | 97.5% CI |
| tP-Au[a] | 31.37 | 15.82 | 43.01 | 31.88 | 16.68 | 43.12 |
| tEA-An-I[a] | 37.77 | 28.87 | 43.11 | 39.54 | 31.67 | 43.37 |
| tASIA_PACIFIC[a] | 47.60 | 43.77 | 51.88 | 46.95 | 43.67 | 51.81 |
| tE_ASIA_PACIFIC[a] | 56.60 | 52.34 | 67.89 | 57.85 | 52.30 | 69.63 |
| tEURASIA_A[a] | 137.68 | 95.77 | 171.43 | 121.38 | 78.51 | 166.98 |
| tN-D[a] | 320.25 | 300.68 | 360.48 | 314.07 | 300.31 | 343.23 |
| tHOMININ[a] | 531.23 | 469.95 | 587.70 | 558.22 | 503.66 | 596.01 |
| t Introgression D- > PACIFIC[a] | 43.12 | 31.97 | 50.33 | 43.10 | 29.38 | 50.29 |
| Introgression D- > PACIFIC | 0.016 | 0.004 | 0.025 | 0.016 | 0.004 | 0.025 |
| t Introgression Xd- > D[a] | 68.02 | 42.92 | 96.54 | 77.90 | 44.63 | 98.51 |
| Introgression ER- > D | 0.011 | 0.002 | 0.025 | 0.013 | 0.003 | 0.029 |
| t HOMININ_ER[a] | 1474.61 | 1025.78 | 1972.40 | 1492.86 | 1021.66 | 1973.87 |
| t Introgression Xe- > ASIA/PACIFIC[a] | 53.73 | 45.75 | 64.68 | 51.03 | 45.01 | 58.04 |
| Introgression Xe- > ASIA/PACIFIC | 0.034 | 0.019 | 0.048 | 0.026 | 0.007 | 0.047 |
| t Introgression N- > OOA[a] | 77.49 | 58.81 | 111.15 | 69.47 | 56.20 | 88.65 |
| Introgression N- > OOA | 0.019 | 0.007 | 0.034 | 0.013 | 0.002 | 0.026 |
| Admixture D- > G | NA | NA | NA | 0.511 | 0.079 | 0.792 |
| t Admixture N;D- > Xe[a] | NA | NA | NA | 304.41 | 211.20 | 375.21 |
| t Xe-D[a] | 279.16 | 206.25 | 359.88 | NA | NA | NA |

**Table 2 Mean and 95% credible interval of the posterior distributions of time and introgression parameters for Model F and Model H**

The parameters were estimated by means of ABC-DL
*NA* Not Applicable, *P* Papua, *Au* Australia, *EA* East Asia, *An* Andaman, *I* India, *A* Africa, *D* Denisova, *N* Altai Neanderthal, *t* time
[a]Kya assuming a generation time of 29 years[38]

Nevertheless, the Bayes factor between Model H and F is so close to one that the difference in the posterior probability of the two models is hardly worth mentioning[39]. Furthermore, in both models ABC-DL parameter estimation suggests that Xe had a similarly long independent evolutionary history (Table 2, Supplementary Table 8, 9 and 10). Therefore, we can only claim with confidence that this population had separated early and close to the Neanderthal–Denisova split time. The consensus of the best models is represented by a trichotomy of the three branches. Given the results from the confusion matrix, supporting a similar posterior probability for identifying by our ABC-DL approach the model that generated the simulation, it is unlikely that this model uncertainty is due to the relatively recent time of divergence that we estimated between Neanderthal and Denisovans. In contrast, this lack of confidence for distinguishing among F and H could be due to a limitation of using a whole-genomic SFS approach, which does not take into account additional information from the spatial distribution within the genome of introgressed regions (i.e. Sprime[20]).

Independently of this limitation, both F and H models can explain some previously contradictory results. For example, they would explain the dearth of African ancestry in some populations (like Andamanese, Indian and Oceanian populations[6] or Tibeto-Burman). The admixed Neanderthal–Denisova archaic ghost population of Model H that introgresses in Asia also explains previous results showing that Asian populations have a higher amount of Neanderthal and Denisovan introgression than Europeans[12,16]. Furthermore, both most supported models agree with the conclusion recently reached by Browning et al.[20] that proposed two Denisovan-like introgressions in Asian and Oceanian populations using a completely independent methodology. However, Browning et al.[20] proposed that Altai Denisovan-like admixed with East Asians whereas the same sister-Denisovan population would have introgressed with Asian and Papuan populations. In our model comparisons we had not considered such model given that East Asian, South and South-East Asia populations showed a similar amount of Denisovan ancestry in our analyses; we considered a shared Denisovan-like ancestry from the same population source as a more parsimonious scenario than the described in[20]. Similarly, Neanderthal dilution in Europeans is also suggested before[40]. In our analyses, the only model considering Neanderthal dilution in Europeans (Model B) is not supported compared to models that consider introgression from unknown hominin. This result does not necessarily exclude Basal Eurasian admixture, as we have not considered more complex models considering Neanderthal dilution in Europeans and archaic introgression in Asian and Oceanians, which we disregarded a priori as less parsimonious than the considered ones. Nevertheless, we would like to point out that the less Neanderthal ancestry in European compared to Asians can be partially explained (if not all) by the Model F/H due to shared ancestry (and thus shared derived alleles) between all EEH populations.

In the same context, there are other demographic models that have not been considered here, since they are not relevant for our conclusions. The exact relationship of Oceanian populations with Eurasian populations is contested. Some studies[7,11] suggest that European and East Asian have more recent ancestry than Oceanian populations. In contrast, other studies argued that East Asian and Oceanian populations have a recent ancestry[6,8]. In the present study we have considered the later model as it is gaining recent support[41,42]. Nevertheless, an early split of Australian and Papuan populations from Eurasians would not have directly affected the percentage of estimated Xn introgression in East Asia and South Asia but only the proportions of Denisovan introgression in Oceania. In fact, we observed that the percentage of

Denisovan introgression in Oceania significantly drop to 1.6% (95% CI 0.4%, 2.5%) in Model H compared to previous estimates[7,8]. However, taking into account the percentage of Xn introgression in Oceania, the total amount of Denisovan-like introgression is 4.2%, within the range of proposed values in Oceania[7,8]. Similarly, it has been suggested that the introgressed EEH populations split off from the Altai and Denisovan samples (that we used as proxies in our analyses) quite early[16]. In principle, this could affect the estimated percentage of Neanderthal introgression out of Africa and the Denisovan introgression in Oceania. Nevertheless, this sample bias cannot explain the relative increase of both Neanderthal and Denisova ancestry in Asian populations compared to Europeans (Table 1 and Fig. 1). The drift, which these EEH populations had faced after their separation of the Neanderthal and Denisovan populations from the proxy samples, would be at random and independent of the relative increase of derived alleles shared between Asian and EEH populations compared to Europeans. This suggests that we need at least one more introgression event to explain this differential increase. Moreover, we have not modelled an ancestral AMH population introgressed into Neanderthals and admixture between Neanderthal and Denisova[14,43], as this would affect all OOA populations equally and will have negligible effect on them, thus not affecting differences between Neanderthal and Denisova ancestry in Asians. Sankararaman et al.[44] proposed a back migration from Oceanian populations to Asia to explain the different proportions of archaic ancestry. However, such a model implies that such Oceanian back migration had a similar genomic impact on populations across a vast region (Indian, Andamanese and East Asian), which seems highly unlikely. Similarly, it has been proposed that the Neanderthal ancestry is diminished over time by differential negative selection acting stronger in Europe. Although it is accepted that Neanderthal ancestry decreases over time in European populations most likely by negative selection[45], it is unclear why this process would not be similar in Asian populations[46]. We also have not inspected a model where a population that is an outgroup of both human and Neanderthal–Denisova lineages has introgressed in Asian populations. A similar population has been hypothesized to have introgressed into Denisovans[14,16], but an introgression from such a population into Asians will not result in the observed Asian excess of Neanderthal and Denisova ancestry compared to Europeans as detected by D-statistics and can be omitted from the analysis.

In conclusion, we have implemented a method that couples ABC with DL that allows to use all the data at the same time and properly comparing complex demographic models. Our novel analytical approach based on ABC-DL supports a model of human evolution in which the OOA populations have had introgression not only from Neanderthals and Denisovans, but also from a third, related group, still genetically and archaeologically uncharacterized. However, the whole genome SFS approach that we used in our ABC-DL implementation does not consider additional information—such as the length and nature of the introgressed fragments—that can be important for distinguishing between some of the models. Further development of the ABC-DL framework will be required to include the fragment length of the ancestry of the genomic fragments. Similarly, new analyses and ancient genome sequences will be required to better understand the nature and consequences of this third related group in the genome of modern humans.

## Methods

**Samples and variant calling**. We used already published and publicly available data from 1000 Genome Project 3rd Phase high coverage data[47], Simons Genome Diversity Project (SGDP)[8] and Mondal et al. 2016[6] (available at European

Nucleotide Archive: PRJEB11455 and PRJEB16019) (see Supplementary Table 1). We downloaded four Europeans (EUR), four East Asians (ASN) and four Africans (AFR) BAM files from 1000 Genome Project; two Papuans and two Australian Aboriginal (PAC) sequences (FASTQ files) from SGDP data set; two Irula, two Birhor (IND); two Jarawa and two Onge (AND) bam files from our previous work[6]. We also downloaded two Neanderthals (NEAN)[13,16,43] and one Denisova (DENI)[12] BAM files from their respective sites. In case the BAM files were not accessible, we mapped the FASTQ files ourselves to create BAM files. All the information on the samples and their respective origin is mentioned in Supplementary Table 1. The number of individuals were kept similar for EUR, ASN, AFR, IND, AND and PAC to avoid unwanted sample size bias for D-statistics, F4-ratio test and ABC analysis.

**D-statistics and F4 ratio test**. We have used the mapability mask as suggested previously[5] thus removing repeated regions (although without this filter the results are same). We have also used two Irulas from SGDP data set to map on the chimpanzee genome reference to exclude any biases from the reference genome (see also Supplementary Methods). We used Admixtools1.4[23] to calculate D-statistics and F4 ratio test. We developed a pipeline (Supplementary Figure 2) using the observed values of D-statistics (EUR, AFR, NEAN, Ancestral), D-statistics (ASN, AFR, NEAN, Ancestral), D-statistics (PAC, AFR, DENI, Ancestral), D-statistics (ASN, EUR, NEAN, Ancestral) and D-statistics (ASN, EUR, DENI, Ancestral) to estimate the percentage of archaic introgression using simulated data generated with ms[48] (Supplementary Table 4) in different simple demographic models (Supplementary Figure 1) while keeping all the other demographic parameters constant (Supplementary Table 3). In case of models with a single introgression parameter estimation, the optimize function from R was used; in case of two parameters *optim* function was used with L-BFGS-B methods with default parameters[49]. Both *optimize* and *optim* functions are iterative methods for solving nonlinear optimization problems. These are used to find better solution for a given function (See Supplementary Note 1 for more details).

**ABC-DL method**. In order to statistically test among competing demographic models and estimating the posterior distributions of the parameters of a given model, we developed an ABC-DL approach using as raw summary statistic the SFS (see also Supplementary Note 2). For each population we considered two samples, except in the case of Neanderthal and Denisovan, where we only considered the Altai Neanderthal and Denisovan individuals. Simulations were generated using FastSimcoal2[21]. Each simulation comprised simulating 9643 genomic regions, comprising 651 Mb, and one diploid individual per population using the demographic topology of one of the proposed trees (see Fig. 2). We used the option *--multiSFS* of FastSimcoal2 to generate the SFS. We considered a generation time of 29 years per generation[38], a uniform recombination rate of 1.0e-8 and a mutation rate of 1.61e-8 ± 0.13e-8[50]. At each simulation and genomic region, we sampled the mutation rate $\mu$ assuming a normal distribution with mean 1.61e-8 and standard deviation of 0.13e-8. However, since the SNP density at each region depends on the fraction of considered fragments out of genes and CgP islands defined in[51] (see Supplementary Note 2 for data preprocessing), we scaled the mutation rate $\mu_r$ of each region $r$ by the fraction of included fragments ($L_c$) over the total length of the region ($L_r$):

$$\mu \sim N(1.61e-8, \ 0.13e-8) \tag{1}$$

$$\mu_r = \mu \frac{L_c}{L_r} \tag{2}$$

In our implementations of the ABC algorithm, we considered an error threshold of 1000/100,000 (i.e. retaining the parameters/models of the 1000 simulations out of 100,000 showing the smallest error with the observed SS). For model comparison, we considered running a multinomial logistic regression on accepted simulations; for parameter estimation, a lineal local regression on accepted simulations. All of these analyses were conducted with the R package abc[52], using the script *postpr* with option *mnlogistic* for model comparison and the script *abc* with option *loclinear* for parameter estimation.

For the DL topology, we generated a supervised four layer feedforward DL network using Encog3.4[53] (see Supplementary Note 2). An innovative aspect of our implementation of DL is the development of SFS-like noise injection. The latest hyper-parameter is motivated by the fact that the DL is trained with data simulated from the proposed models of interest, whereas the real model that generated the observed genomic data is -by definition- more complex and (at best) partially overlaps/generalizes with (some) of the considered models. In this case, it can be expected that a trained DL with data from models different from the real one will produce biased SS-DL with regards to real data. In our study, we propose overcoming this problem by applying an innovative modified version of the noise injection algorithm[4]. In the classical injection algorithm, white noise (i.e. from a normal distribution with mean 0 and a given standard deviation) is dynamically added at each iteration to the elements of the training data set in order to force model generalization. However, properly adding white noise requires independence between the input variables, which is not the case of the considered SS. In order to perform a noise injection-like approach in the case of the SFS statistic, we took

advantage that we have two individuals for each population—with the exception of Neanderthal and Denisovan. We split the observed data set in two, one comprising samples that will be used to add "observed data-like" noise to the simulated SFS, and a second one that will be used for the estimation of the posterior distributions using the ABC approach. At each iteration of the DL training algorithm, we define for the $j$ cell of the SFS of simulation $i$ from model $s$:

$$\text{SFS}_{i,j}^{s} = (1 - \alpha_i)\text{SFS}_{i,j}^{s} + \alpha_i \text{SFS}_{i,j}^{r} \tag{3}$$

where $\alpha$ is sampled from a uniform distribution in the range [0,0.2] and $r$ corresponds to the observed "real" data. Introducing noise from an independent observed data set offers an additional advantage. By analogy with image detection, we can imagine that the simulated training data corresponds to a sharp high-resolution image of an object from the known simulated model/parameter, whereas the observed $r$ data is at best an image showing one of the model/parameter objects at low resolution on a foggy environment (i.e. the parts of the model generating the data not considered in our models). This difference between the data used for training and the final observed data that we want to classify can introduce biases in the final model classification. By generating new SFS patterns by means of noise injection, increasing robustness in the network classification against model departures can be expected.

**Code availability**. We implemented a JAVA-based pipeline available at https://github.com/oscarlao/ABC_DL to (i) generate FastSimcoal2 input files from the prior distributions of each parameter and read the SFS from the simulation, (ii) implement feedforward DL using the JAVA Encog3.4[53] and (iii) generate input files for ABC analyses with the R package abc[52]. Details of the full implementation of the DL and ABC-DL algorithm are provided in the Supplementary Note 2.

**Reporting Summary**. Further information on experimental design is available in the Nature Research Reporting Summary linked to this article.

## Data availability
All the sequence data came from their respective publicly available data sources and all other downstream data analysis are available on request to the corresponding authors.

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

## Acknowledgements

We thank Iago Maceda Porto for useful comments on the manual and testing of the JAVA suite. M.M was supported by the European Union through the European Regional Development Fund (Project No. 2014-2020.4.01.16-0030). For J.B, this study has been possible thanks to grant BFU2016-77961-P (AEI/FEDER, UE) awarded by the Agencia Estatal de Investigación (MINECO, Spain) and with the support of Secretaria d'Universitats i Recerca del Departament d'Economia i Coneixement de la Generalitat de Catalunya (GRC 2017 SGR 702). Part of the "Unidad de Excelencia María de Maeztu", funded by the MINECO (ref: MDM-2014-0370). O.L. was supported by a Ramón y Cajal grant from the Spanish Ministerio de Economia y Competitividad (MINECO) with reference RYC-2013-14797, a BFU2015-68759-P (MINECO/FEDER) grant and the support of Secretaria d'Universitats i Recerca del Departament d'Economia i Coneixement de la Generalitat de Catalunya (GRC 2017 SGR 937). O.L. also acknowledges the Spanish Ministry of Economy, Industry and Competitiveness (MEIC) to the EMBL partnership; Centro de Excelencia Severo Ochoa; CERCA Programme/Generalitat de Catalunya; the Spanish Ministry of Economy, Industry and Competitiveness (MEIC) through the Instituto de Salud Carlos III; Generalitat de Catalunya through Departament de Salut and Departament d'Empresa i Coneixement; the co-financing by the Spanish Ministry of Economy, Industry and Competitiveness (MEIC) with funds from the European Regional Development Fund (ERDF) corresponding to the 2014-2020 Smart Growth Operating Program.

## Author contributions

M.M., J.B. and O.L. conceived the study. M.M. and O.L. performed analyses. M.M., J.B. and O.L. wrote the manuscript.
