## [Peer Review File · Nature Communications]

Reviewers' comments:

Reviewer #1 (Remarks to the Author):

The manuscript has been revised to my satisfaction. I have no further comments.

Reviewer #2 (Remarks to the Author):

I'm basically happy with the revision by Mondal et al, especially since they pointed out that one of my major critiques (re: dilution models) was a result of my misreading the manuscript. I do appreciate the authors attempt to clarify the point better in the manuscript, which I think reads more clearly.

I'm sympathetic to the authors comment that the specific topology of the Eurasian populations probably doesn't impact their results super strongly, and I'm okay with that.

However, upon re-reading the manuscript, it struck me that the signal for an unsampled archaic population, particularly one closely related to Denisovans, isn't that strange—in fact, we know that the introgressing Denisovan population was pretty diverged from the Altai Denisovan for which we have a reference genome. Moreover, it's unclear to me how exactly the authors combine the information from the Altai and Vinjia high coverage Neandertals, which are not just fairly diverged from each other but also show differential similarity to the introgressing Neandertal population (with Vindijia being much closer).

Because of this, I'm really not surprised by their results at all. In fact, I think that sort of explains the support for Model H: the Xe branch is basically accounting for things that look a bit like the reference Neandertals but are a bit diverged and look a bit like the reference Denisovan but are a bit diverged.

Because of this, I think it might be important for the authors to test one additional model: where the introgression doesn't come directly from N (which should probably be represented by the Vindijia Neandertal) and D, but instead from populations that split off from those particular samples. Essentially, I'm suggesting combining model E and F, but not allowing introgression to come directly from the N and D branch. My intuition is that this model would be very hard to tell apart from model H (modulo, perhaps, additional Denisova in East Asia ala Browning et al) but is more in line with things we already think.

I prefer to sign my reviews. My name is Joshua Schraiber.

Reviewer #3 (Remarks to the Author):

The manuscript "Approximate Bayesian computation with a Deep Learning algorithm (ABC-DL) supports a third archaic introgression common to all Asian and Oceanian human populations" improved greatly over a previous version of the same work "An overall vision of ancient introgression in the human lineage out of Africa", addressing very well most of the reviewers' criticisms.

First of all, I agree with other reviewers that this work provides an interesting methodology, but that the old title was too grandiose. I think the new title is much better, although I definitely find important that the authors would stress the major advantages of their approach, one of which is that their method allows a global treatment of the problem, "an overall vision" as they previously defined. I am also happy that the discussion and conclusions about the relationship of this "third archaic hominin"

with the Neandertal and Denisovan are tuned down. All this, contribute to make the paper more digestible.

Regarding previous comments, I appreciate the authors' replies in clarifying the differences between their "explorative simulations" looking at D-stats and f_4 , and their ABC-DL looking at the SFS. I feel that also the text improved in clarity as a consequence. However, my point was that neglecting local genomic information and only focusing at global information (regardless if that is SFS, D or f_4) can be a choice reasonable from the computational point of view, but that can lead to an important loss of information and resolution. And this loss of resolution is pretty apparent in the higher ability of Browning et al. in discriminating between Denisova-related ancestry or a third 50% Neandertal 50% Denisovan hominin lineage. The authors are now clearer about this, and I appreciate it. However, the rationale they suggest for not taking into account local genomic information is only valid for their own methodological framework as they state "...Decomposing this SFS by fragment would increase the number of input cells to a non-affordable number of inputs for the DL; this is not possible with the present state of computation possibilities...". Therefore, if the authors want to present a paper that is focused on the methods, they should make clear why the reader should choose their method over others. This means being clear about pros and cons.

I acknowledge that everything is already written in the paper in different parts. But maybe something could be rephrased to say clearly:

- "use our method, it is cool because you can treat all data at the same time and properly compare models!"

- "But beware, because there is additional information that you are clearly neglecting, and from our very same case study it is clear that this can be decisive to discriminate between some of the models".

Besides, if the authors want anyone to be able to use their method, this should be documented much better. Not even better, it should just be documented. I tried to open the link provided in the text, and I simply could not find any documentation or good README file. A method with a code that nobody can use is not very useful. Also, I wanted to test the code a bit, but without a good documentation that is impossible. It should be mandatory that the authors provide a good documentation before publication.

Regarding the model tested I understand the rationale for not including Browning's model explicitly. This paper came out shortly before the submission of the first version of this manuscript, and before that as the authors state "In our model comparisons we had not considered such model given that East Asian, South and South-East Asia populations showed a similar amount of Denisovan ancestry in our analyses; we considered a shared Denisovan-like ancestry from the same population source as a more parsimonious scenario than the described in 20". However, now that paper is out. As far as I can empathize with the authors for the bad timing, the model that might have looked non-parsimonious before the publication of Browning et al., it is now the starting hypothesis, the "most parsimonious". So now one could try to read a bit more into the tested models taking into account the findings of Browning et al. In fact, I appreciate much more the current manuscript because this is already done much more than in the previous version. However, I think the authors should do even a bit more.

-From the point of view of the discussion, for example, the authors show the posterior distribution of the parameters for model H, and one sees no difference in N_e for Neandertals and Denisovans, with Neandertals possibly even a tiny bit higher (that by the way, it is not what observed in the data). What is observed for model D? Does one sees higher N_e for Denisovans?

-Also, isn't model F pretty much the same as Browning et al's model? If the authors now say "The Bayes factor between model H and model F is so close to 1 that difference in the posterior probability of the two models is hardly worth mentioning" why in the main text only model H deserves its own figure? I think that this is misleading. The authors should provide graphical representations of both

models. The reality (as it seems also from the comments of other reviewers) is that most likely Browning's et al model, and model F, are correct. So it is great that the authors developed a new method, and even without looking at local genomic information they get to conclusions that are pretty similar to those of Browning et al. (they could even brag about it more in the paper). But still, a distracted reader looking at the main figure should not get a misleading biological message. If the authors really do not want to duplicate figure 3 in two parts for model H and model F, the authors could modify a bit fig3 to represent both models, changing a bit the trifurcation in something that can represent both models and write that explicitly in the caption.

I thank the authors for the plots and clarification that the authors generated to answer to some of my previous doubts, and I agree with them that this is a minor point that could be neglected as uninteresting for the general reader.

The authors state in the discussion:

"Although it is accepted that Neanderthal 327 ancestry is diluted over time in European populations most likely by negative selection 46, it is unclear why this process would not be similar in Asian populations 47". This sentence is correct in its main message, but I find confusing the association of the term "dilution" with selection. I think usually "dilution" is used more in connection with the Neanderthal-poor contribution of Basal Eurasians (Lazaridis et al., 2014, 2016) than to selection. This is not only a semantic point: the authors do not discuss at all the question of why there is more Neanderthal-like haplotypes in Asian populations. I think this is due to the fact that the authors do not find evidence for two Neanderthal admixtures (one additional in East Asians) and concurrently do not address the dilution hypothesis from Basal Eurasians. I suggest the authors to comment briefly about this, since the benefit of this paper is exactly addressing all admixture events with archaics at once.

A final minor point is about the estimates of Neanderthal-Denisovan split. The authors suggest that the differences could be explained by different mutation rates. This is false. In this paper the estimate of the Neanderthal and Denisovan split is about half the time that of humans-archaics.

Rogers et al. infer that the Neanderthal and Denisovan, and modern human and archaics separated more or less at the same time (~300 generations apart). This is an extreme scenario, already criticised by Mafessoni and Pruefer 2017, and plausibly the time of separation of Neanderthal and Denisovans was relatively more recent, similarly to what estimated in the present paper. But also the estimates of Pruefer et al., 2014, 2017 indicate that the split of Neanderthal and Denisovan was not as recent. This is a minor point, but if drawn, one should be honest and not suggest that mutation rates explain these differences, when they don't - the difference is in the relative time of the split, not in the absolute dates.

Unfortunately I think that this point has to be made, since the reduced divergence between Denisovan and Neanderthals expected here could have some effects and make one expect that Neanderthal and Denisovan introgressed SNPS should be more similar than they are actually are. Would this be able to hide the existence of a second Neanderthal admixture into east Asians? Probably not, but this discrepancy should still be mentioned so that readers can also be more aware of the differences of the models inferred here in respect to previously inferred demographic models. Especially since the divergence between Neanderthal and Denisovan likely play an important role in distinguishing between different archaic admixture events.

Reviewer #2 (Remarks to the Author):

I'm basically happy with the revision by Mondal et al, especially since they pointed out that one of my major critiques (re: dilution models) was a result of my misreading the manuscript. I do appreciate the authors attempt to clarify the point better in the manuscript, which I think reads more clearly. I'm sympathetic to the authors comment that the specific topology of the Eurasian populations probably doesn't impact their results super strongly, and I'm okay with that.

We thank the reviewer for this positive reply to our re-submission.

However, upon re-reading the manuscript, it struck me that the signal for an unsampled archaic population, particularly one closely related to Denisovans, isn't that strange—in fact, we know that the introgressing Denisovan population was pretty diverged from the Altai Denisovan for which we have a reference genome. Moreover, it's unclear to me how exactly the authors combine the information from the Altai and Vindija high coverage Neandertals, which are not just fairly diverged from each other but also show differential similarity to the introgressing Neandertal population (with Vindija being much closer).

Reply to reviewer: all the ABC-DL analyses were conducted with the Altai genome, not with Vindija.

We explained this in the supplementary material:

“We considered the individuals for ABC-DL analyses: Altai, BIR-08, BIR-11, Denisova, ERS1358131, ERS1358132, ERS1042142, ERS1042181, HG00096, HG00419, HG00759, HG01500, HG02922, HG03052, IL-01, IL-04, JAR-27, JAR-32, ONG-1, ONG-12, NA12891, NA12892, NA18525, NA18939, NA19238 and NA19239 (Table S01).”

We have included this information also in the main text to avoid any confusion with Dstats, where we also use the Vindija sample:

Line 392: “For each population we considered two samples, except in the case of Neanderthal and Denisovan, where we only considered the Altai Neanderthal and Denisovan individuals.”

The F4ratio test (Figure 1) was devised such a way that the introgression can happen from a population, which is closer to Vindija compared to Altai. Prufer et al, 2014 Supplementary Section 14 has a discussion on this topic. On that paper, Figure S14.2 and Table S14.7 was used for “Other Neanderthal” where the other Neanderthal (in our case that is Vindija) is closer to the introgressed Neanderthal population than the Altai Neanderthal. We have used same F4ratio test from that paper.

Independently of this fact, if all OOA populations have a single introgression event from a single

Neanderthal population, the Dstat results will equally affect the European and East Asian rather than producing the differential results reported in Table 1. Despite it can change the absolute value of introgression depending on use of Altai or Vindija, the relative value of introgression ($EUR-ASN \approx 0$) between European and East Asian will remain same regardless of Altai or Vindija.

Because of this, I'm really not surprised by their results at all. In fact, I think that sort of explains the support for Model H: the Xe branch is basically accounting for things that look a bit like the reference Neanderthals but are a bit diverged and look a bit like the reference Denisovan but are a bit diverged. Because of this, I think it might be important for the authors to test one additional model: where the introgression doesn't come directly from N (which should probably be represented by the Vindija Neanderthal) and D, but instead from populations that split off from those particular samples. Essentially, I'm suggesting combining model E and F, but not allowing introgression to come directly from the N and D branch. My intuition is that this model would be very hard to tell apart from model H (modulo, perhaps, additional Denisova in East Asia ala Browning et al) but is more in line with things we already think.

Reply to reviewer: we agree with reviewer #2 that this model would produce a similar genomic SFS than the one from model H. In fact, they would be the identical when i) the effective population size of the population sister to the Neanderthals and the population sister to the Denisovans is similar to the effective population size of the admixed ghost population of model H, ii) the introgression of these sister populations in Asia take place at the same time as in model H and iii) with introgression proportions similar to the admixture proportions in the ghost population of model H. We can think many other models that combine Neanderthal and Denisovan ancestry to produce a similar SFS than in model H. Because this is theoretically expected, there is no way the proposed ABC-DL (which focuses on whole genomic SFS) can distinguish between these models. Henceforth, we do not think is really needed to empirically test this particular proposed model, but to acknowledge the limitation of the method in the manuscript (also in agreement with reviewer #3). From our point of view, the only way we see to solve this problem would be by looking at a local level, not at a genomic level. According to model H, introgressed fragments should be a mixture of Neanderthal and Denisova ancestry. According to the proposed model by the reviewer, some fragments would be Neanderthal-like, some Denisovan-like. Unfortunately, this approach would require considering independently the SFS of thousands of fragments, which is, to the best of our knowledge, computationally unfeasible using dense feedforward NN. It is likely that adapting other NN structures, such as convolutional neural networks, could fix this problem. However, this goes far beyond the original purpose of this work, which was to show that ABC coupled to DL can help identifying scenarios suggesting archaic introgressions.

We already pointed the limitation of our method in the previous version of the manuscript.

“this lack of confidence for distinguishing among F and H could be due to a limitation of using a whole-genomic SFS approach, which does not take into account additional information from the spatial distribution within the genome of introgressed regions.”

We have extended this sentence to account for other models such as the proposed by the reviewer whose SFS genomic signature would be indistinguishable from model H.

In line 285: *“For example, a model that considers that the archaic introgression in Asian populations occurred from a Neanderthal-like and a Denisovan-like population could produce a genomic SFS like the model H. The only difference between both models would be that, according to model H, introgressed fragments should be a mixture of Neanderthal-like and Denisova-like ancestry. In contrast, according to this other model, some fragments would be Neanderthal-like, some Denisovan-like.”*

We have also modified the title of the article to indicate the limitation of our results:

Approximate Bayesian computation with Deep Learning algorithm (ABC-DL) supports at least a third archaic introgression common to all Asian and Pacific human populations.

Finally, we have written a final statement explicitly indicating the limitation of this methodology (as well as all the strong points, which in our opinion overrule this drawback):

Line 350

“However, the whole genome SFS approach that we used in our ABC-DL implementation does not consider additional information –such as the length and nature of the introgressed fragments- that can be important for distinguishing between some of the models. Further development of the ABC-DL framework will be required to include the fragment length of the ancestry of the genomic fragments. Similarly, new analyses will be required to better understand the nature and consequences of this third related group in the genome of modern humans”

Our results is somewhat similar to Browning et al. with slight differences, we have dedicated a paragraph for this on the discussion section:

Line 302: Furthermore, both most supported models agree with the conclusion recently reached by Browning et al. ²⁰ that proposed two Denisovan-like introgressions in Asian and Oceanian populations using a completely independent methodology.

Reviewer #3 (Remarks to the Author):

The manuscript "Approximate Bayesian computation with a Deep Learning algorithm (ABC-DL) supports a third archaic introgression common to all Asian and Oceanian human populations" improved greatly over a previous version of the same work "An overall vision of ancient introgression in the human lineage out of Africa", addressing very well most of the reviewers' criticisms. First of all, I agree with other reviewers that this work provides an interesting methodology, but that the old title was too grandiose. I think the new title is much better, although I definitely find important that the authors would stress the major advantages of their approach, one of which is that their method allows a global treatment of the problem, "an overall vision" as they previously defined. I am also happy that the discussion and conclusion..".about the relationship of this "third archaic hominin" with the Neandertal and Denisovan are tuned down. All this, contribute to make the paper more digestible.

Regarding previous comments, I appreciate the authors' replies in clarifying the differences between their "explorative simulations" looking at D-stats and f4, and their ABC-DL looking at the SFS. I feel that also the text improved in clarity as a consequence. However, my point was that neglecting local genomic information and only focusing at global information (regardless if that is SFS, D or f4) can be a choice reasonable from the computational point of view, but that can lead to an important loss of information and resolution. And this loss of resolution is pretty apparent in the higher ability of Browning et al. in discriminating between Denisova-related ancestry or a third 50% Neandertal 50% Denisovan hominin lineage. The authors are now clearer about this, and I appreciate it. However, the rationale they suggest for not taking into account local genomic information is only valid for their own methodological framework as they state "..Decomposing this SFS by fragment would increase the number of input cells to a non-affordable number of inputs for the DL; this is not possible with the present state of computation possibilities..".

Therefore, if the authors want to present a paper that is focused on the methods, they should make clear why the reader should choose their method over others. This means being clear about pros and cons. I acknowledge that everything is already written in the paper in different parts. But maybe something could be rephrased to say clearly:

- "use our method, it is cool because you can treat all data at the same time and properly compare models!"

- "But beware, because there is additional information that you are clearly neglecting, and from our very same case study it is clear that this can be decisive to discriminate between some of the models".

Reply to reviewer: We have included a sentence in Line 345 explicitly explaining the pros and cons of

the current implementation of the ABC-DL framework:

“In conclusion, we have implemented a method that couples ABC with DL that allows using all data at the same time and properly comparing complex demographic models. Our novel analytical approach based on ABC-DL supports a model of human evolution in which the OOA populations have had introgression not only from Neanderthals and Denisovans, but also from a third, related group, still genetically and archaeologically uncharacterized. However, the whole genome SFS approach that we used in our ABC-DL implementation does not consider additional information – such as the length and nature of the introgressed fragments- that can be important for distinguishing between some of the models. Further development of the ABC-DL framework will be required to include the fragment length of the ancestry of the genomic fragments.”

Besides, if the authors want anyone to be able to use their method, this should be documented much better. Not even better, it should just be documented. Also, I wanted to test the code a bit, but without a good documentation that is impossible. It should be mandatory that the authors provide a good documentation before publication.

Reply to reviewer: We totally agree with the reviewer, and we apologize for not having properly provided a clear documentation of the methodology. We have generated a pdf manual describing all the steps required to implement and run demographic models and to conduct the ABC-DL approach that we introduced in our manuscript. Furthermore, we have produced a Javadoc of the main classes and methods that are used in the implemented approach.

Finally, we describe step by step how to implement in JAVA a model comparison and parameter estimation using two demographic models, one of them includes archaic introgression. Some people from the lab not working in this project read the manual and used the java code in the github to reproduce results. However, using this pipeline requires moderate to mastering JAVA programming skills, which obviously we cannot provide in the Manual.

Regarding the model tested I understand the rationale for not including Browning's model explicitly. This paper came out shortly before the submission of the first version of this manuscript, and before that as the authors state "In our model comparisons we had not considered such model given that East Asian, South and South-East Asia populations showed a similar amount of Denisovan ancestry in our analyses; we considered a shared Denisovan-like ancestry from the same population source as a more parsimonious scenario than the described in 20". However, now that paper is out. As far as I can empathize with the authors for the bad timing, the model that might have looked non-parsimonious before the publication of Browning et al., it is now the starting hypothesis, the "most parsimonious". So now one could try to read a bit more into the tested models taking into account the findings of

Browning et al. In fact, I appreciate much more the current manuscript because this is already done much more than in the previous version. However, I think the authors should do even a bit more.

-From the point of view of the discussion, for example, the authors show the posterior distribution of the parameters for model H, and one sees no difference in Ne for Neandertals and Denisovans, with Neandertals possibly even a tiny bit higher (that by the way, it is not what observer in the data). What is observed for model D? Does one sees higher Ne for Denisovans?

Reply to reviewer: The estimated effective population size of Neanderthal and Denisovan from the posterior distribution is similar in all models, being the mean of the Ne of Neanderthals ~1.2 times the Ne of Denisovan. However, we would like to point out that we are reporting the mean value of the distribution. Nevertheless, the distribution of the Ne of the Neanderthal is not completely unimodal.

-Also, isn't model F pretty much the same as Browning et al's model? If the authors now say "The Bayes factor between model H and model F is so close to 1 that difference in the posterior probability of the two models is hardly worth mentioning" why in the main text only model H deserves its own figure? I think that this is misleading. The authors should provide graphical representations of both models. The reality (as it seems also from the comments of other reviewers) is that most likely Browning's et al model, and model F, are correct. So it is great that the authors developed a new method, and even without looking at local genomic information they get to conclusions that are pretty similar to those of Browning et al. (they could even brag about it more in the paper). But still, a distracted reader looking at the main figure should not get a misleading biological message. If the authors really do not want to duplicate figure 3 in two parts for model H and model F, the authors could modify a bit fig3 to represent both models, changing a bit the trifurcation in something that can represent both models and write that explicitly in the caption.

Reply to reviewer: We have included in Figure 3 a graphical representation of model F and modified the main text introducing it. Similarly, we have updated Table 2 including main findings of model F. We have also included in the supplementary material (Table S09) the mean and credible interval of the posterior distribution of the parameters estimated with model F.

The authors state in the discussion:

"Although it is accepted that Neanderthal ancestry is diluted over time in European populations most likely by negative selection, it is unclear why this process would not be similar in Asian populations". This sentence is correct in its main message, but I find confusing the association of the term "dilution" with selection. I think usually "dilution" is used more in connection with the Neanderthal-poor contribution of Basal Eurasians (Lazaridis et al.,2014,2016) than to selection.

Reply to the reviewer: we agree with the reviewer that we have not used properly the term “dilution”, which can be easily misunderstood with the archaic dilution to which the reviewer refers. The real meaning is “diminished over time” (line 337).

This is not only a semantic point: the authors do not discuss at all the question of why there is more Neandertal-like haplotypes in Asian populations. I think this is due to the fact that the authors do not find evidence for two Neandertal admixtures (one additional in East Asians) and concurrently do not address the dilution hypothesis from Basal Eurasians. I suggest the authors to comment briefly about this, since the benefit of this paper is exactly addressing all admixture events with archaics at once.

Reply to the reviewer: In line 303 of the previous manuscript we wrote that “Model H also explains previous results showing that Asian populations have a higher amount of Neanderthal and Denisovan introgression than Europeans.” Without making explicit acknowledgement of the fact that this is because of the admixed Neanderthal-Denisova “ghost” archaic population. We have rewritten it to explain it better:

Line 294: *“The admixed Neanderthal-Denisova archaic ghost population of model H that introgresses in Asia also explains previous results showing that Asian populations have a higher amount of Neanderthal and Denisovan introgression than Europeans.”*

In the previous version of the manuscript, we also referred to the dilution in Europe, indicating that (Line 307).

“In our analyses, the only model considering Neanderthal dilution in Europeans (Model B) is not supported compared to models that consider introgression from unknown hominin. This result does not necessarily exclude Basal Eurasian admixture, as we have not considered more complex models considering Neanderthal dilution in Europeans and archaic introgression in Asian and Oceanians, which we disregarded a priori as less parsimonious than the considered ones.

We added this additional line (Line 316) to make it more clear:

“Nevertheless, we would like to point out that the less Neanderthal ancestry in European compared to Asians can be partially explained (if not all) by the Model F/H due to shared ancestry (and thus shared derived alleles) between all EEH populations.”

A final minor point is about the estimates of Neandertal-Denisovan split. The authors suggest that the differences could be explained by different mutation rates. This is false. In this paper the estimate of the Neandertal and Denisovan split is about half the time that of humans-archaics. Rogers et al. infer that the Neandertal and Denisovan, and modern human and archaics separated more or less at the same time (~300 generations apart). This is an extreme scenario, already criticised by Mafessoni and Pruefer 2017, and plausibly the time of separation of Neandertal and

Denisovans was relatively more recent, similarly to what estimated in the present paper. But also the estimates of Pruefer et al.,2014,2017 indicate that the split of Neandertal and Denisovan was not as recent. This is a minor point, but if drawn, one should be honest and not suggest that mutation rates explain these differences, when they don't - the difference is in the relative time of the split, not in the absolute dates.

Unfortunately I think that this point has to be made, since the reduced divergence between Denisovan and Neandertals expected here could have some effects and make one expect that Neandertal and Denisovan introgressed SNPS should be more similar than they are actually are. Would this be able to hide the existence of a second Neandertal admixture into east Asians? Probably not, but this discrepancy should still be mentioned so that readers can also be more aware of the differences of the models inferred here in respect to previously inferred demographic models. Especially since the divergence between Neandertal and Denisovan likely play an important role in distinguishing between different archaic admixture events.

Reply to reviewer: We are thankful to reviewer to identify this point. When we said mutation rate can explain we meant that it can explain the discrepancy between our parameter of divergence between Neanderthal and Denisova (~320 kya) and what is proposed by Prufer et al (~400 kya). To put it very simply if we scale 320 kya with lower mutation rate that was used in Prufer et al, it will give $320 \times (1.6/1.45) \sim 355$ kya (333-389 kya CI interval percentile), we think that the result is similar between us and Prufer et al (as we have not modeled ancient genome explicitly). However, it is certainly different of what is proposed by Roger et al when we analyzed it in deep. As reviewer suggested, after further inspection we realized that, in all of our cases, the time divergence of AMH and EEH (558 kya), is close to double of Neanderthal and Denisova split (320 kya). Showing at least in our model indeed Neanderthal and Denisova shared ancestry between them for considerable amount of time unlikely what is proposed by Roger et al. We have updated the line by removing the 502 kya with a mutation rate of $1e-08$ by generation. We did not go much further details about why we are not supporting Roger et al. as this paper is more directed towards introgression in modern humans rather than ancient effective population size of EEH populations. :

“The estimated divergence between Denisovan and Neanderthals is relatively recent compared to the estimated in 39 (502 kya with a mutation rate of $1e-08$ by generation) and 16 (381–473 kya when considering a mutation rate of $1.45e-08$ per generation) but higher than in 16 (190–236 kya when considering a mutation rate of $2.9e-08$ per generation). Since the number of non-shared variants depends on the time of divergence, the genetic drift at each population and the mutation rate, a possible explanation for the differences in time of divergence could be the mutation rate, which substantially variate among studies.”

REVIEWERS' COMMENTS:

Reviewer #2 (Remarks to the Author):

I am happy with the authors revisions, modulo a misunderstanding about my concerns about the modeling. I had a direct communication with the authors who assured me that they would address the concern in a minor revision, and I would be happy to accept the paper after that is addressed. Just to be specific, I would like the authors to comment on the relevance of a model where we simply failed to sample the actual introgressing lineages (which is certainly the case), and how that would impact their results. I am happy to communicate with the authors before resubmission to ensure that we are on the same page about what I mean.

I prefer to sign my reviews. My name is Joshua Schraiber.

Reviewer #3 (Remarks to the Author):

I am very satisfied with the this final version of the manuscript: the authors are very clear and honest about their results and its limitations. Although biologically one could consider this a confirmatory result, it is methodologically very important to see what can be inferred without using local genomic information. So I certainly recommend the manuscript for publication.

My only suggestion would be in the final published version of the paper to write more explicitly the authors' findings in the abstract, compatibly with the length of the abstract. I think it would be nice to be able to see already in the abstract what this "third introgression event" can be - either an ancestral hominin related to the Denisova-Neandertal branch, most likely more related to Denisovans than Neandertals.

Reviewer #2 (Remarks to the Author):

I am happy with the authors revisions, modulo a misunderstanding about my concerns about the modeling. I had a direct communication with the authors who assured me that they would address the concern in a minor revision, and I would be happy to accept the paper after that is addressed. Just to be specific, I would like the authors to comment on the relevance of a model where we simply failed to sample the actual introgressing lineages (which is certainly the case), and how that would impact their results. I am happy to communicate with the authors before resubmission to ensure that we are on the same page about what I mean.

After contacting Reviewer 2, he explained to us that he was referring to the model proposed by Prufer et al, 2014, in which the Neanderthal and Denisovan introgression from our basic model (and hence extended to all the other models) came from two populations that split off the Altai and Denisovan populations.

Reply to reviewer: We agree with the Reviewer 2 that the Altai and Denisovan samples that were used as proxy of the archaic populations are unlikely to be the true source populations of introgression. We agree that this can have a putative bias effect in the estimated proportions of introgressions when using these two populations. This is apparent when using Vindija for D-statistics calculation instead of Altai Neanderthal: we see increase of Neanderthal introgression amount in all the populations out of Africa. However, this increase is proportional to all the populations out of Africa. Therefore, the relative increase of Neanderthal introgression amount in all Asian populations compared to European populations is independent of which Neanderthal sample we use. The same is true for the Denisova introgression in Oceania. Introgression of Denisova in Oceanian populations in principle cannot explain the increase of Denisova ancestry in all Asian populations compared to Europeans. This is because Neanderthal and Denisova tree is highly diverged from modern human tree (separated ~700kya). Therefore, even if there was a split and random drift from the last 200 ky for the sequenced hominin and the true introgressed hominin, it cannot explain the statistical increase of Neanderthal and Denisova in Asian populations compared to European populations, which just separated 50 kya. Therefore, the signal that we recovered with the ABC-DL cannot be an artifact due to the archaic samples that we used as proxies.

We have explained it in the main text (line 313):

“Similarly, it has been suggested that the introgressed EEH populations split off from the Altai and Denisovan samples (that we used as proxies in our analyses) quite early¹⁶. In principle, this could affect the estimated percentage of Neanderthal introgression out of Africa and the Denisovan introgression in Oceania. Nevertheless, this sample bias cannot explain the relative increase of both Neanderthal and Denisova ancestry in Asian populations compared to Europeans (Table 1 and Figure 1). The drift, which these EEH populations had faced after their separation of the Neanderthal and Denisovan populations from the proxy samples, would be at random and independent of the relative increase of derived alleles shared between Asian and EEH populations compared to Europeans. This suggests that we need at least one more introgression event to explain this differential increase.”

Reviewer #3 (Remarks to the Author):

I am very satisfied with the this final version of the manuscript: the authors are very clear and honest about their results and its limitations. Although biologically one could consider this a confirmatory result, it is methodologically very important to see what can be inferred without using local genomic information. So I certainly recommend the manuscript for publication.

My only suggestion would be in the final published version of the paper to write more explicitly the authors' findings in the abstract, compatibly with the length of the abstract. I think it would be nice to be able to see already in the abstract what this "third introgression event" can be - either an ancestral hominin related to the Denisova-Neandertal branch, most likely more related to Denisovans than Neandertals.

Reply to reviewer: We have added a part of line as the reviewer suggested.